# 3D Component Segmentation Network and Dataset for Non-Cooperative Spacecraft

**Guangyuan Zhao** [1,2,3] , **Xue Wan** [2,3,*] , **Yaolin Tian** [1,2,3] , **Yadong Shao** [1,2,3] **and Shengyang Li** [2,3]

1 School of Computer Science and Technology, University of Chinese Academy of Sciences, Beijing 100049, China; zhaoguangyuan19@mails.ucas.ac.cn (G.Z.); tianyaolin21@mails.ucas.ac.cn (Y.T.); shaoyadong18@csu.ac.cn (Y.S.)
2 Technology and Engineering Center for Space Utilization, Chinese Academy of Sciences, Beijing 100094, China; shyli@csu.ac.cn
3 Key Laboratory of Space Utilization, Chinese Academy of Sciences, Beijing 100094, China
* Correspondence: wanxue@csu.ac.cn

**Abstract:** Spacecraft component segmentation is one of the key technologies which enables autonomous navigation and manipulation for non-cooperative spacecraft in OOS (On-Orbit Service). While most of the studies on spacecraft component segmentation are based on 2D image segmentation, this paper proposes spacecraft component segmentation methods based on 3D point clouds. Firstly, we propose a multi-source 3D spacecraft component segmentation dataset, including point clouds from lidar and VisualSFM (Visual Structure From Motion). Then, an improved PointNet++based 3D component segmentation network named 3DSatNet is proposed with a new geometrical-aware FE (Feature Extraction) layers and a new loss function to tackle the data imbalance problem which means the points number of different components differ greatly, and the density distribution of point cloud is not uniform. Moreover, when the partial prior point clouds of the target spacecraft are known, we propose a 3DSatNet-Reg network by adding a Teaser-based 3D point clouds registration module to 3DSatNet to obtain higher component segmentation accuracy. Experiments carried out on our proposed dataset demonstrate that the proposed 3DSatNet achieves 1.9% higher instance mIoU than PointNet++_SSG, and the highest IoU for antenna in both lidar point clouds and visual point clouds compared with the popular networks. Furthermore, our algorithm has been deployed on an embedded AI computing device Nvidia Jetson TX2 which has the potential to be used on orbit with a processing speed of 0.228 s per point cloud with 20,000 points.

**Keywords:** non-cooperative spacecraft; 3D spacecraft component segmentation network; 3D spacecraft component segmentation dataset; registration; deep learning

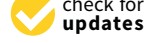


## 1. Introduction

The OOS of spacecraft has already become one of the hot study areas in aerospace science over the past decades [1], particularly on fault detection approaches, influence of disturbances, modeling errors, and various uncertainties in the real systems [2]. Most existing OOS missions, such as DEOS (Deutsche Orbital Servicing Mission) [3], iBOSS (Intelligent Building Blocks for On-Orbit Satellite Servicing) [4], and SIS (Space Infrastructure Servicing) [5], are based on cooperative spacecraft, which are equipped with cooperative marks and can communicate with the servicing spacecraft [6]. However, more general cases in OOS are aimed at non-cooperative spacecraft. For instance, the OOS tasks can be repairing a malfunctioned satellite that has no cooperative mark and communication ability. Without cooperative marks and sometimes even no prior knowledge of the target spacecraft, the OSS task for non-cooperative spacecraft is still an open research area facing many technical challenges [7].

The component-level segmentation of non-cooperative spacecraft is the premise and core technology to achieve the more complex navigation-related tasks of the OOS task.



Most existing spacecraft recognition approaches are based on 2D images [8]. Machine learning, particularly deep learning, is being increasing utilised in space applications, mirroring the groundbreaking success in many earthbound problems [9]. Several datasets, such as Spacecraft-Parts [10] and SolarPanelDataset [11], have been published to evaluate the performance of deep learning algorithms in spacecraft 2D component segmentation. In [12], a hybrid convolution network, which used a two-stage target detection network and Mask R-CNN [13] segmentation network, was used to join the global features and partial semantic information for spacecraft component recognition. L Zhang et al. [14] proposed a real-time docking ring detection method for on-orbit spacecraft based on arc extraction and ellipse parameters estimation. The experiments were conducted on a novel SPARK dataset [15]. A non-cooperative image recognition network was constructed in [16], in which two networks were used to identify non-cooperative spacecraft panels. Visual images, however, are easily be affected by illumination conditions, so the image-based methods failed to perform well when spacecraft was in the shadow. Moreover, view angles of 2D images are restricted, thus some components may be occluded from the current perspective of view.

Compared to 2D-based methods, the advantages of using 3D point cloud for spacecraft component segmentation is listed as follows. Firstly, the 3D point clouds contain the global structure of the target spacecraft without the problem of occlusion [17]. Moreover, some components of spacecraft can be easily recognized via a 3D point cloud. As spacecraft is an artificial object, there contain many regular planes and lines, which are more suitable for 3D-based recognition. For example, the solar panel appears to be quite flat in 3D point cloud, while this is hard to be described in 2D images.

However, to the best of our knowledge, no 3D component segmentation network has been proposed for non-cooperative spacecraft yet. Point cloud part segmentation is a topic that has been studied for several years in the field of computer vision, which has recently regained significant attention due to the success of deep learning. These approaches can be divided into three categories: MLP-Based, Graph-Based, and Convolution-Based methods [18]. The graph-based part segmentation methods use graph structure [19–21] to reflect the topographic relationship between 3D points, however, the size of these models are too large to be used in OOS. Similarly, the convolution-based component segmentation approaches [22–24] also require large computation and memory resources, which are not suitable for OOS missions. One of the typical MLP-based algorithms is PointNet [25], which processed 3D point clouds with shared multi-layer perception and uses max pooling to extract global features. Though efficient and effective, PointNet failed to extract local features, limiting its ability to recognize fine-grained spacecraft components. PointNet++ [26], which added a farthest point sampling layer and a ball query layer before PointNet was proposed to extract both global and local features.

Although MLP-based methods have the potential to be used in 3D spacecraft component segmentation tasks, there are still some problems that remain unsolved. Firstly, the 3D models of spacecraft are rare and have different shapes. To the best of our konwledge, no 3D spacecraft component segmentation dataset is available yet. Moreover, the data distribution between different spacecraft components can be largely varied. For example, for a spacecraft made up of 2500 points, a panel occupies 1000–1800 points, while an antenna only occupies 50–100 points. This imbalanced data distribution may lead to unreliable segmentation results for those components with fewer points. Furthermore, memory, power, compute, and reliability have to be taken into account for deep learning methods used on orbit. the largest limiting factor is the limited computing ability which is not suitable for large-scale networks deployment. Finally, some spacecraft are customized productions, it is impossible to include all the components in the dataset, so these kinds of special components cannot be identified through the network.

To tackle the above problems, we proposed a 3D spacecraft component segmentation network named 3DSatNet. Our contributions can be summarized as follows:

1. A multi-source dataset for 3D spacecraft component segmentation is proposed. The dataset includes both visual point clouds and lidar point clouds using 3D reconstruction and 3D scanning. Finally, the dataset covers 32 3D spacecraft models including 3 component categories: body, panel, antenna.

2. We peoposed a 3D spacecraft component segmentation network named 3DSatNet based on PointNet++ and made some improvements A geometrical-aware feature extraction layer named FE layer using ISS (Intrinsic Shape Signatures) [27] keypoint detection methods is proposed to extract geometric and transformation invariant features. Furthermore, a new weighted cross-entropy loss is proposed to increase the weight of components with fewer points, such as antenna. We fine tune the 3DSatNet by freezing the feature extractor and retraining the last full connected layers. We made some model pruning methods on PointNet++ to reduce the model size and compute power demand.

3. When the partial prior point clouds of the target spacecraft are known, we proposed a 3DSatNet-Reg network by adding a Teaser-based [28] 3D point cloud registration module behind 3DSatNet to obtain higher component segmentation accuracy. For the purpose of achieving semantic corrections, we assigned the label of the prior point clouds to the closest point on semantic point clouds generated by 3DSatNet.

The remaining sections in this paper are organized as follows. Section 2 detailed presents the 3D multi-source spacecraft segmentation dataset and the architecture of the proposed 3DSatNet Network. Section 3 presents the experimental results. The result is discussed in Section 4. This paper is concluded in Section 5.

## 2. Materials and Methods

As shown in Figure 1, this section describes the 3D spacecraft component segmentation materials and methods we proposed. The 3DSatNet is trained on earth using our 3D spacecraft component segmentation dataset. The input of the 3DSatNet network is the 3D point cloud of the target spacecraft which is reconstructed from a sequence of on-orbit visual images or lidar-SLAM. The 3D point cloud is transferred into semantic point clouds via the 3DSatNet. When the partial prior point clouds of the spacecraft are known, 3D point cloud registration module based on Teaser [28] is added to 3DSatNet to obtain higher component segmentation accuracy, which is named as 3DSatNet-Reg.

Section 2.1 is the introduction of the 3D spacecraft component segmentation dataset we proposed to train the 3DSatNet model. Section 2.2 detailedly introduces the proposed 3D component segmentation network named 3DSatNet, and Section 2.3 presents the network 3DSatNet-Reg for accurate segmentation.

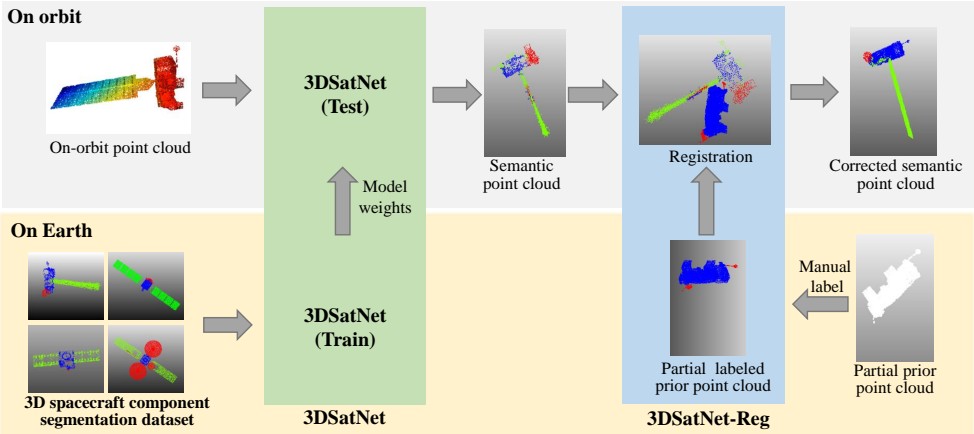

**Figure 1.** The pipeline of the proposed 3D spacecraft component segmentation system.

### 2.1. Dataset

We download the CAD model of the published spacecraft from open-source websites, such as NASA https://www.nasa.gov/, accessed on 21 March 2022 and Free3d https://free3d.com/, accessed on 21 March 2022. Lately, we print the CAD models utilizing the 3D printing technology to get the 3D printed models. As shown in Figure 2, Figure 2a illustrates the CAD model of AcrimSAT https://nasa3d.arc.nasa.gov/detail/jpl-vtad-acrimsat, accessed on 21 March 2022 downloaded from NASA, and Figure 2b presents the 3D printed model of this satellite. 28 CAD models and 5 3D printed models of different sizes shapes are eventually obtained (Considering the expensive cost of the 3D printing technology, we only print a trivial number of spacecraft CAD models).

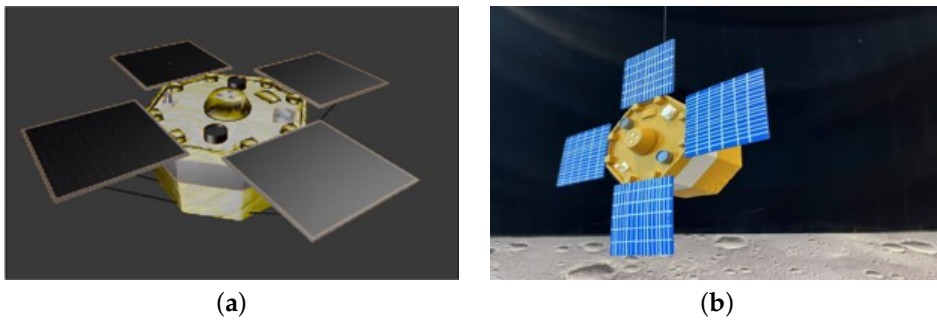

| (**a**) | (**b**) |

**Figure 2.** The CAD model and 3D printed model of ArcimSAT: (**a**) CAD model. (**b**) 3D printed model.

Further, the simulated point clouds and the actual point clouds are obtained. There are three methods to gain 3D point clouds: generating 3D point clouds via AirSim, generating 3D point clouds via VisualSFM, generating 3D point clouds via 3D scanning of spacecraft models.

Generation of 3D point clouds via AriSimAirSim is an open-source and cross-platform simulation engine which can imitate the lidar data for multiple different channels. In experiments, lidars of 32 channels are configured on the simulation engine, and parameters are listed in the following Table 1.

**Table 1.** The configuration of the simulation lidar in AirSim.

| Parameters Name | Parameters |
| --- | --- |
| Number Of channels | 32 |
| Rotations per second | 10 |
| Points per second | 100,000 |
| Horizontal FOV range | (0, 360) |
| Vertical FOV range | (−20, 20) |
| $xyz$ | (0, 0, −1) |
| Roll Pitch Yaw | (0, 0, 1) |

To obtain a rather completed 3D model, the flight trajectory of the simulation engine is designed in Figure 3a. Besides, to obtain accurate scanning results, it is necessary to rebuild the collision model to the max for each component imported into the spacecraft. The final scanned 3D point cloud is shown in Figure 3c.

Generation of 3D point clouds via VisualSFM: VisualSFM [29] is a GUI application for 3D reconstruction using 2D images. VisualSFM is able to run very fast by exploiting multicore parallelism in feature detection, feature matching, and bundle adjustment. There are two methods to gain image sequences: Blender simulation, or using the camera on the Nvidia Jetson TX2 https://developer.nvidia.com/embedded/jetson-tx2, accessed on 15 March 2022.

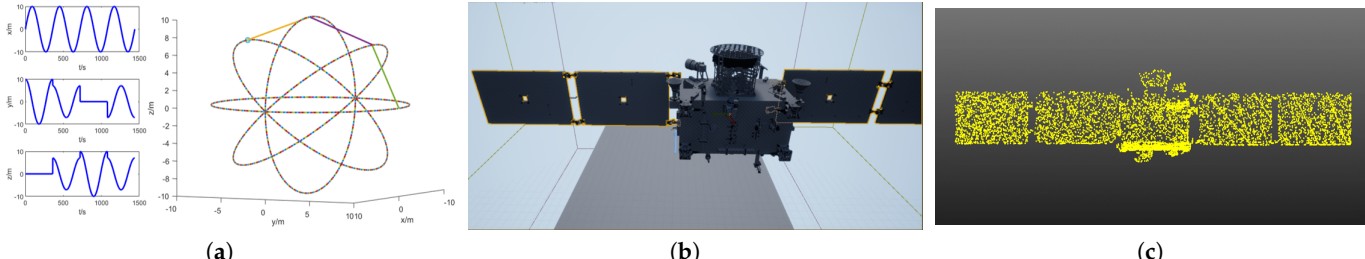

| (a) | (b) | (c) |

**Figure 3.** Airsim simulation system: (**a**) Airsim simulated trajectory. (**b**) Airsim simulated trajectory. (**c**) Airsim simulated point cloud.

Blender2.8.2 https://www.blender.org/, accessed on 15 March 2022 is used to gain the simulation image sequence of spacecraft. As shown in Figure 4a, the CAD model of the Aqua satellite is imported into Blender, and a circle with a radius of 10 m at 10 m above the CAD model is drawn. Setting the camera's light center aligned with the model, 600 simulation images with the size of $1024 \times 1024$ are obtained by shooting around along the circular trajectory. Lately, the simulated vision point clouds are obtained via VisualSFM, and the noise within them is manually removed to get the relatively clean point clouds as shown in Figure 4b.

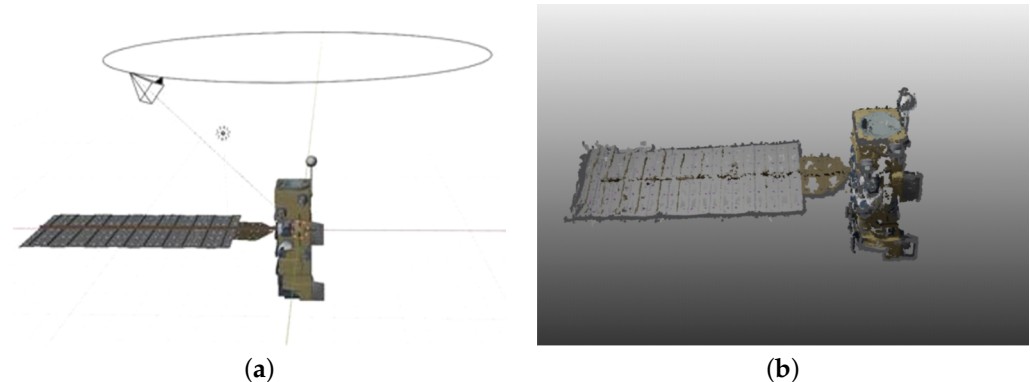

| (a) | (b) |

**Figure 4.** Blender simulation system: (**a**) Blender simulation platform. (**b**) Simulation visual point cloud.

The acquisition of actual image sequences uses the camera on the Nvidia Jetson TX2 developer component https://developer.nvidia.com/embedded/jetson-tx2, accessed on 15 March 2022, as shown in Figure 5a. The image acquisition frequency is 30 FPS, The TX2's camera is aligned to the 3D printed model, 600 actual images with the size of $1920 \times 1080$ are obtained by shooting around along the circular trajectory. The real vision point clouds are obtained using the VisualSFM method.

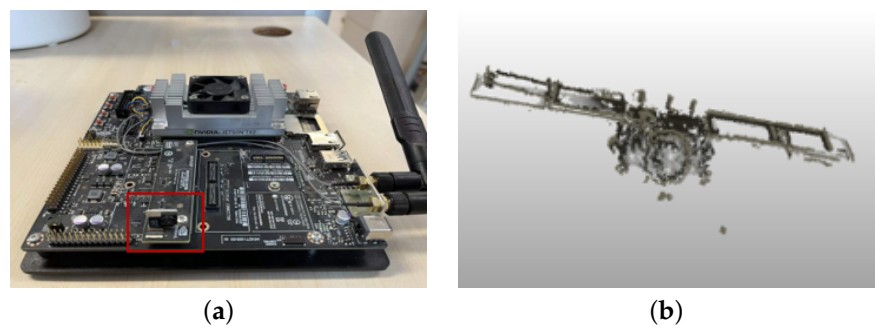

| (a) | (b) |

**Figure 5.** TX2 simulation system: (**a**) Camera on TX2. (**b**) Actual visual point cloud.

Generation of 3D point clouds via 3D scanner: The scanner selected in this experiment is EinScan Pro 2X Plus Handheld 3D scanner https://www.einscan.com/handheld-3d-scanner/2x-plus/, accessed on 17 March 2022, as shown in Figure 6a below. Figure 6b shows the scene of the 3D scanner scans a 3D printing model of spacecraft. According to the different materials of spacecraft, the scanning mode includes handheld fine scanning and handheld fast scanning. The handheld fine scanning mode is used to scan high reflective objects by pasting control points on the target, while the handheld fast scanning mode which does not need to paste control points is feasible for scanning general materials.

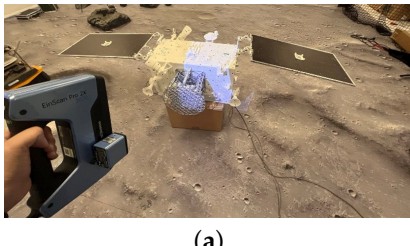 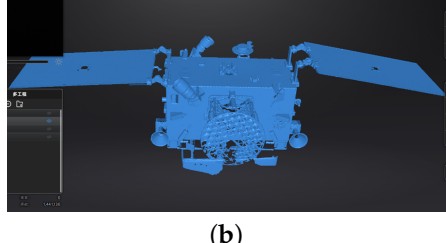

(**a**)　　　　　　　　　　　　　　　　　　　　　　　　　　　(**b**)

**Figure 6.** 3D scan system: (**a**) 3D scanner scan spacecraft. (**b**) 3D scan point clouds.

As the initial CAD models have different sizes and shapes, varying from around 50 cm to 6 m, which makes it difficult for MLP-based models to deal with. Furthermore, different CAD models have different coordinate systems, for example, AcrimSAT satellite may use the left vertex of the panel as the origin of coordinates while LRO https://nasa3d.arc.nasa.gov/detail/jpl-vtad-lro, accessed on 23 April 2022 may use the centroids. Thus, point cloud pre-processing methods including size normalization and centroid-based centralization is required.

The pre-processing methods of point clouds are based on Equations (1) and (2), after this pre-processing, the point clouds are transformed within the range of (0, 1) in $x$, $y$, and $z$-axis. Where $P \subset \mathbb{R}^{m \times 3}$ is raw point clouds, $m$ is point numbers, $P'$ is the centralized and normalized point clouds data.

For a point cloud $P \subset \mathbb{R}^{m \times 3}$ containing $n$ points, $P_i$ represents the $i$th points in $P$, $P'_i$ represents the $i$th point in the point cloud after centralization. $P_{morm}$ represents the point cloud after normalization. $P'_{\min}$ represents the minimum value of the three channels $(x, y, z)$ in $P'_i$. Similarly, $P'_{\max}$ represents the maximum value of the three channels. Through the above two steps, the coordinate range of the spacecraft is normalized to (0, 1), and the coordinate origin is changed to the center of mass of the spacecraft.

$$P'_i = P_i - \frac{1}{n} \sum_{i=1}^{n} P_i \tag{1}$$

$$P_{norm} = \frac{P' - P'_{\min}}{P'_{\max} - P'_{\min}} \tag{2}$$

In this paper, our annotation system is based on open-source software CloudCompare https://www.cloudcompare.org/, accessed on 15 March 2022, and Figure 7 presents its details.

As shown in Table 2, the 3D spacecraft component segmentation dataset divides spacecraft into three categories: body, panel, antenna. In this dataset, the points distribution between different components in a spacecraft can be largely varied, a solar panel occupies 8000–15,000 points, while an antenna only occupies 300–1500 points in point clouds with 20,000 points.

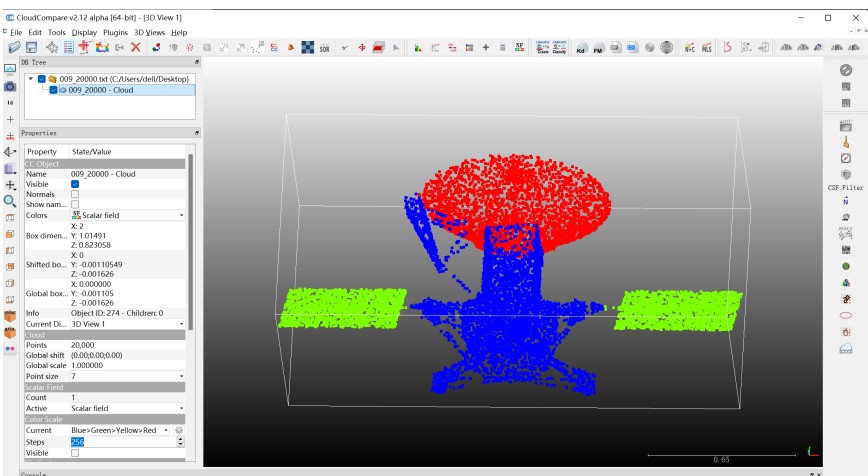

**Figure 7.** Screenshot of annotation system.

**Table 2.** Label of components in 3D spacecraft component segmentation dataset.

| Components | Label |
|---|---|
| Body | 0 |
| Panel | 1 |
| Antenna | 2 |

The results of some annotations are shown in Figure 8.

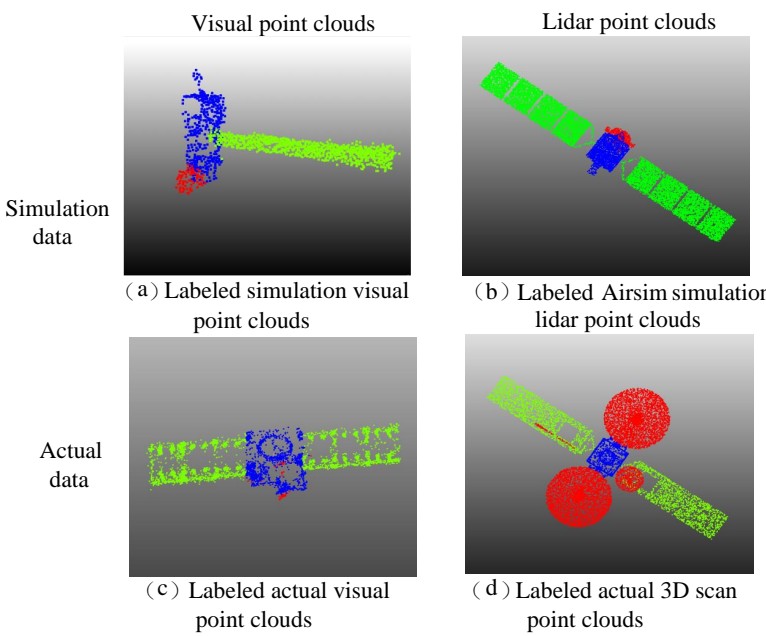

Visual point clouds | Lidar point clouds

Simulation data

（a）Labeled simulation visual point clouds | （b）Labeled Airsim simulation lidar point clouds

Actual data

（c）Labeled actual visual point clouds | （d）Labeled actual 3D scan point clouds

**Figure 8.** Visualization of the labeled spacecraft. (Red: antenna, Blue: body, Green: panel).

As shown in Figure 9, the format of the annotated point clouds is described as follows: the first three columns are $x, y, z$ coordinates, the following three columns are normal vectors, and the last column is semantic labels.

A total of 66 raw point clouds of four types are generated, as is shown in Figure 3 each containing 20,000 points. Then, various methods are adopted to enhance the obtained point clouds, such as randomly adding noise points, translating and rotating point clouds, and adding different degrees of occlusion. After data enhancement, 792 point clouds are obtained.

| | 1 | 2 | 3 | 4 | 5 | 6 | 7 |
|---|---|---|---|---|---|---|---|
| | X coord. X | Y coord. Y | Z coord. Z | Nx | Ny | Nz | Scalar |
| | 0.087939 | 0.176172 | 0.112132 | 0.673712 | 0.515784 | 0.529225 | 1.000000 |
| | 0.146260 | 0.199795 | 0.106410 | 0.000978 | 0.000978 | 0.999999 | 1.000000 |
| | 0.088806 | 0.180350 | 0.108030 | 0.590884 | 0.657839 | 0.467016 | 2.000000 |
| | 0.091920 | 0.446349 | 0.112413 | 0.000978 | 0.000978 | 0.999999 | 1.000000 |
| | 0.091467 | 0.372710 | 0.117116 | 0.191268 | 0.981537 | 0.001145 | 1.000000 |
| | 0.094285 | 0.271121 | 0.117605 | 0.102575 | 0.057487 | 0.993063 | 2.000000 |
| | 0.089145 | 0.462416 | 0.110965 | 0.085931 | 0.741341 | 0.665605 | 1.000000 |
| | 0.142843 | 0.471528 | 0.107089 | 0.001139 | 0.183394 | 0.983039 | 1.000000 |

**Figure 9.** Visualizationof the labeled data formation.

**Table 3.** Summary of the proposed 3D spacecraft component segmentation dataset.

| | Simulated Visual | Actual Visual | Simulated Lidar | Actual Lidar |
|---|---|---|---|---|
| Acquisition methods | Visual SFM | Visual SFM | AirSim | 3D scan |
| Source | CAD models | 3D printed spacecraft | CAD models | 3D printed spacecraft |
| Model Types | 28 | 5 | 28 | 5 |
| Point Cloud Number [1] | 336 | 60 | 336 | 60 |

[1] The point cloud number is the enhanced point cloud number and the summary of the two versions.

To facilitate the testing of various models, the final released dataset consists of two versions: one with 20,000 points per point cloud which provides a detailed representation of the three-dimensional structure of spacecraft, the other with around 2500 points per point clouds which share similar point numbers per shape as the ShapeNet dataset. To be specific, each point cloud with 20,000 points is downsampled to around 2500 points using ISS methods.

### 2.2. 3DSatNet

The proposed 3DSatNet is inspired by PointNet++ [26]. The pipeline of the proposed 3DSatNet is demonstrated in Figure 10. Firstly, A geometrical-aware feature extraction layer named the FE layer is proposed to extract geometric and transformation invariant features. When extracting features, the point cloud is continuously down-sampled. Then, the FP (Feature Propagation) layer is used for feature aggregation, which includes an up-sampling and skip connection parts. Moreover, a new loss function is proposed to increase the segmentation accuracy of the component with fewer points, such as antenna. The following sections will describe more details of the network.

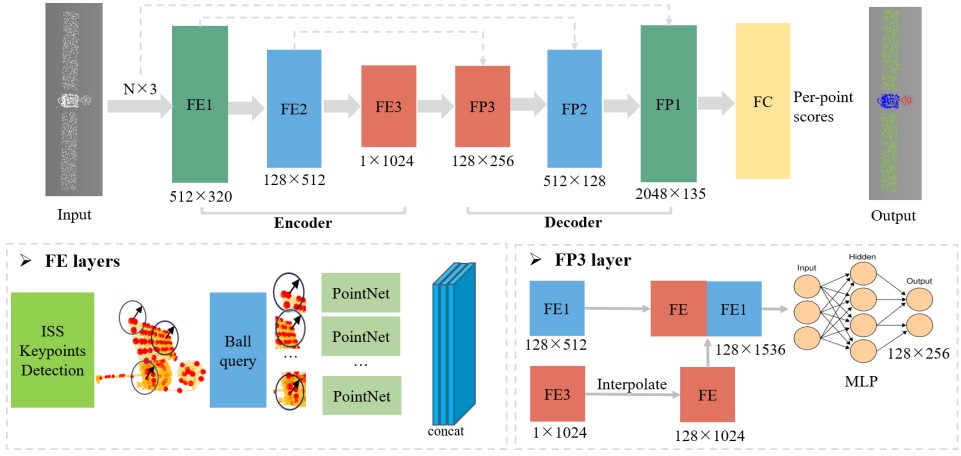

**Figure 10.** The architecture of the proposed 3DSatNet.

### 2.2.1. The Proposed Geometrical-Aware FE Layers

The proposed FE layers contain three parts: ISS (Intrinsic Shape Signatures) point selection, ball query, and PointNet. We first partition the set of points into overlapping local regions by the ISS [27], ball query, and then feed them into PointNet for further feature extractions.

ISS [27] point selection method has better coverage of the structure point set given the same number of centroids. The ISS saliency measure is based on the Eigenvalue Decomposition (EVD) of the scatter matrix $S(p)$.

$$S(p) = \frac{1}{N} \sum_{q \in \mathcal{N}(p)} (q - \mu_p)(q - \mu_p)^T$$
$$\text{with } \mu_p = \frac{1}{N} \sum_{q \in \mathcal{N}(p)} q \tag{3}$$

$\mathcal{N}(p)$ means the neighborhood point of the point $p$, $S(p)$ means the scatter matrix of the points belonging to the neighborhood of $p$. $q$ represents the points in $\mathcal{N}(p)$. $N$ means the total number of $\mathcal{N}(p)$.

Given $S(p)$, the eigenvalues $\lambda_1, \lambda_2, \lambda_3 (\lambda_1 > \lambda_2 > \lambda_3)$ are computed, points whose ratio between two successive eigenvalues is below a threshold are retained:

$$\frac{\lambda_2(p)}{\lambda_1(p)} < \gamma_{12} \wedge \frac{\lambda_3(p)}{\lambda_2(p)} < \gamma_{23} \tag{4}$$

where $\gamma_{12} = 0.5$ and $\gamma_{23} = 0.5$.

After the detection step, a point will be considered as an ISS keypoint if it has the maximum salience value in a given neighborhood, as is shown in Figure 11.

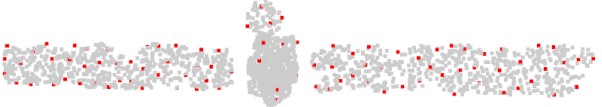

**Figure 11.** The ISS keypoint extraction result. The red points are the selected ISS keypoints.

After obtaining the ISS key points, we take them as the center of the sphere and draw a sphere with a certain radius (given that point clouds in the dataset have been normalized, the radius is taken as 0.2 in this paper). Then the points located in the sphere are indexed. Although the number of points varies across groups, the PointNet layer can convert them into feature vectors with fixed-length.

The PointNet network is used as the backbone to extract feature vectors of point clouds. In order to reduce the model size, this paper prunes the PointNet network structure. For the input point clouds, PointNet firstly uses a T-Net network to learn $3 \times 3$ matrices to convert the input of different perspectives into the input of the same perspective, but this part doubles the size of the network. So we remove the T-Net Network and use MLP to extract transformation invariant features.

The following Figure 12 shows the original structure of PointNet network and the PointNet network structure used in our FE layer.

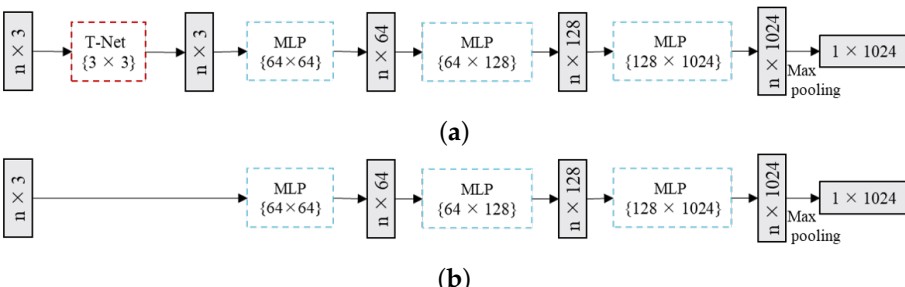

**Figure 12.** Comparison of PointNet structure and the PointNet network structure used in our FE layer: (**a**) Partical structure of initial PointNet. (**b**) Partical structure of PointNet used in our FE layer.

We convert the MLP to matrix form, it can be found that the hidden feature vector is as follows:

$$
h(P) = \begin{bmatrix} x_1 & y_1 & z_1 \\ x_2 & y_2 & z_2 \\ \vdots & \vdots & \vdots \\ x_n & y_n & z_n \end{bmatrix} \begin{bmatrix} w_{11} & w_{12} & \cdots & w_{1k} \\ w_{21} & w_{22} & \cdots & w_{2k} \\ w_{31} & w_{32} & \cdots & w_{3k} \end{bmatrix} + \begin{bmatrix} b_1 & b_2 & \cdots & b_k \end{bmatrix} \tag{5}
$$

$(x_i, y_i, z_i)$ are coordinates of one point $P_i$ in the point cloud $P$, $k$ represents the output chanel of MLP, in our network $k = 64$.

The learnable parameters of the MLP can act as a many rotation martix. The output of the T-Net is a $3 \times 3$ rotation matrix. Considering that the point cloud is put into a $3 \times 64$ MLP, the 64 output channels is stroung enough to make the network's rotation invariant approximately.

### 2.2.2. The FP Layers

FP layers share a similar structure as PointNet++'s decoder part, which adopts a hierarchical propagation strategy with distance-based interpolation and across-level skip links. Taking the FP3 layer as an example, for the $1 \times 1024$ feature output by FE3 layer, it was simply repeated 128 times to $128 \times 1024$. Then the feature was combined with the $128 \times 896$ features output by the FE2 layer to form a new $128 \times (1024 + 896)$ feature, and then mapped to $128 \times 256$ by MLP.

### 2.2.3. Loss Function

As stated in Section 1 that data distribution between different components in a space-craft can be largely varied. To improve the segmentation accuracy of components with few points(e.g., antenna), we proposed a new weighted cross-entropy loss.

$$
L = -\sum_{c=0}^{3} w_c y_c \log(p_c) \tag{6}
$$

$w_c = \frac{N - N_c}{N}$ is a weighted parameter indicating that the point belongs to a certain category, $N_c$ represents the number of points belonging to the class $c$, $y_c$ is a one-hot vector. If the segmentation result is consistent with the prior label, the value of $y_c$ is set as 1; otherwise, the value will be set as 0. $p_c$ represents the probability that the predicted sample belongs to class $c$.

### 2.3. 3DSatNet-Reg

In OOS tasks, several pictures and several frames of lidar scanning results of the target spacecraft can be obtained, and then visual 3D reconstruction technology and lidar 3D reconstruction techniques are used to obtain partial prior point clouds. Aiming at the problems of spatial target component segmentation with prior point clouds, this section

proposes a Teaser-based [28] component semantic segmentation network 3DSatNet-Reg. The labels of the prior point clouds labeled on earth are assigned to the closest point of the raw semantic point cloud gained by 3DSatNet. The pipeline of the proposed 3DSatNet-Reg is shown in Figure 13.

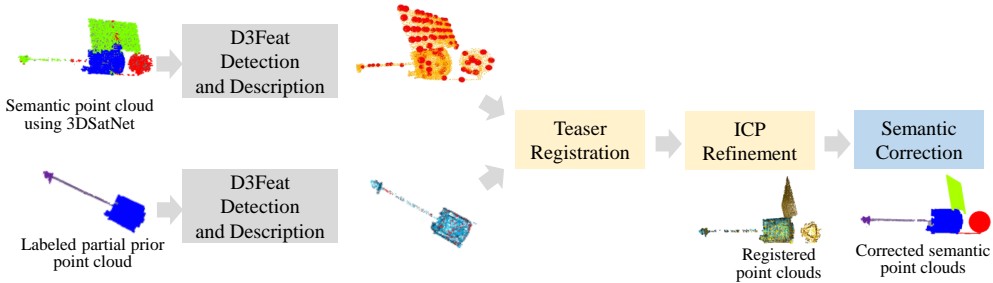

**Figure 13.** Pipeline of the proposed 3DSatNet-Reg network.

The point cloud registration problem aims to find the optimal rotation, translation and scale transformation matrices of the labeled prior point clouds $A = \{a_i\}_{i=1}^{N}$ and the on-orbit semantic point cloud $B = \{b_i\}_{i=1}^{N}$. Considering the influence of the noise of the on-orbit semantic point cloud, the problem can be modeled as Equation (7):

$$b_i = s^\circ R^\circ a_i + t^\circ + o_i + \epsilon_i \tag{7}$$

where $s^\circ > 0$, $R^\circ \in SO(3)$, $t^\circ \in \mathbb{R}^3$, $\epsilon_i$ models the measurement noise, and $o_i$ is a vector of zeros if the pair $(a_i, b_i)$ is an inlier. $\|\epsilon_i\| \leq \beta_i$, where $\beta_i$ is a given bound.

The critical technology to realize partial prior point clouds and overall semantic point cloud registration is to detect the corresponding point pairs on the two point clouds. Traditional feature extraction methods such as FPFH (Fast Point Feature Histogram) and VFH (Viewpoint Feature Histogram) can't perform well, so we use D3Feat [30] to realize local feature detection and description which can be used to find key points and point correspondences with noise.

ICP (Iterative Closest Point) [31] is a commonly used registration algorithm when the initial pose between the two point clouds is not too large. Considering the large poses and scale difference between the prior point clouds and the semantic point cloud, a highly robust Teaser [28] registration algorithm is added to achieve a rough registration and estimation scale of two point clouds before ICP. Then, ICP is used to optimize the registration results.

Teaser [28] is a registration algorithm that can tolerate a large amount of noise. Teaser uses a TLS (Truncated Least Squares) cost that makes the estimation insensitive to a large fraction of spurious correspondences. It assumed the noise is unknown but bounded and proposed a general approach to decouple the estimation of scale, translation, and rotation.

The TRIMS (Translation and Rotation Invariant Measurements) [28] is used to estimate the scale $\hat{s}$.

$$\hat{s} = \arg\min_{s} \sum_{k=1}^{K} \min\left(\frac{(s - s_k)^2}{\alpha_k^2}, \bar{c}^2\right) \tag{8}$$

Equation (8) can be solved exactly and in polynomial time via adaptive voting. bounded noise $\left|\epsilon_{ij}^s\right| \leq \alpha_{ij}$.

Given the scale estimate $\hat{s}$ produced by the scale estimation and TIMS (Translation Invariant Measurements). The rotation estimate $\hat{R}$ is defined as Equation (9) which can be solved in polynomial time via a tight semi-definite relaxation. bounded noise $\|\epsilon_{ij}\| \leq \delta_{ij}$.

$$\hat{R} = \arg\min_{R \in SO(3)} \sum_{k=1}^{K} \min\left(\frac{\left\|\overline{b}_k - \hat{s}R\overline{a}_k\right\|^2}{\delta_k^2}, \bar{c}^2\right) \tag{9}$$

After obtaining the scale and rotation estimates $\hat{s}$ and $\hat{R}$, $\hat{t}$ is estimated from $(a_i, b_i)$ in Equation (10) which can be solved exactly and in polynomial time via adaptive voting.

$$\hat{t}_j = \arg\min_{t_j} \sum_{i=1}^{N} \min\left( \frac{\left( t_j - \left[ b_i - \hat{s}\hat{R}a_i \right]_j \right)^2}{\beta_i^2}, \bar{c}^2 \right) \tag{10}$$

Finally, the ICP [31] algorithm is used to complete fine registration. Estimate the combination of rotation and translation using a point to point distance metric minimization technique defined in Equation (11) which will best align each source point to its match found in the previous step. Then transform the labeled prior point clouds using the obtained scale, translation, and rotation.

$$(R,t) = \underset{R \in SO(3), t \in \mathbb{R}^3}{\arg\min} \sum_{i=1}^{n} \|Ra_i + t - b_i\|^2 \tag{11}$$

After registration, we calculate the closest point between the on-orbit semantic point cloud and the labeled prior point clouds and assign the label to the closest point.

## 3. Results

### 3.1. Overall Performance

This section demonstrates the proposed 3DSatNet on both the ShapeNet-Part dataset and the proposed spacecraft component segmentation dataset. Then, we compare it with SOTA (State-of-the-Art) models.

### 3.1.1. Overall Performance on the ShapeNet-Part Dataset

The ShapeNet-Part dataset consists of 16,881 3D objects, covering 16 shape categories. Most of the point cloud instances are annotated with less than six labels, and there exist 50 parts labels in total. Following the official split policy announced by [32], the dataset is split into training, validation, and testing parts, containing 12,137, 1870, and 2874 shapes respectively. Both the object categories and the object components within the categories are highly imbalanced in the ShapeNet-Part dataset.

In this section, 3DSatNet is compared with the popular networks, including CNN-based, Graph-based, Transformer-based, and MLP-based methods. All of the networks are trained and tested on Nvidia RTX 3090 https://www.nvidia.cn/geforce/graphics-cards/30-series/rtx-3090-3090ti/, accessed on 22 April 2022 with 24 GB memory. We count the GPU occupancy of these networks. In terms of the metric for evaluation, we adopt the instance average mIoU and the class mIoU (mean Intersection-over-Union). The IoU of the shape is calculated by the mean value of IoUs of all components in that category, and instance mIoU is the average of IoUs for all testing shapes. The class mIoU is the average of IoUs for all testing classes. In the table, techniques like PointNet++, SO-Net also exploit normals besides point coordinates as the input features.

mIoU derives from the confusion matrix. Given that there exist $\tau + 1$ classes in spacecraft point clouds (Body: 0, Panel: 1, Antenna: 2), $p_{ij}$ denotes the amount of points of class $i$ inferred to class $j$. Assuming that $p_{ij}$ is true positive, so $p_{ji}$ and $p_{ii}$ represent false positives and true negatives respectively. Under the circumstances, we can derive the mIoU as Equation (12):

$$mIoU = \frac{1}{\tau + 1} \sum_{1}^{\tau} \frac{p_{ii}}{\sum_{j=0}^{\tau} p_{ij} + \sum_{j=0}^{\tau} p_{ji} - p_{ii}} \tag{12}$$

It can be seen in Table 4 that although KPconv [23] achieves the highest class mIoU and Instance mIoU, the large model size and params numbers make it unsuitable for the OOS tasks. The Transformer-based point transformer occupied the most GPU memory, which can't be deployed on Nvidia TX2 (the fastest, most power-efficient embedded AI computing device that has the potential to be used on-orbit). The proposed 3DSatNet not only achieves

the similar instance mIoU and class mIoU as SpiderCNN [33] and DGCNN [34] but also achieves an increase of 0.6% compared to the initial PointNet++_SSG and 0.2% compared to PointNet++_MSG. Particularly, our 3DSatNet achieves the smallest GPU usage (equal to PointNet) because we remove the T-Net part of PointNet and use the geometrical-aware FE layers.

**Table 4.** Comparisons with the SOTA methods on the ShapeNet-Part dataset and 3D spacecraft component segmentation dataset (%).

| Models | Backbone | ShapeNet-Part Dataset | | | Our Dataset | | | |
| | | GPU-Memory | Class mIoU | Instance mIoU | Size | Params | Forward Time | Instance mIoU |
|---|---|---|---|---|---|---|---|---|
| KPconv [23] | CNN | 2.8 GB | 85.1 | 86.4 | 56 MB | 14.2 M | - | - |
| SpiderCNN [33] | CNN | 6.5 GB | 82.4 | 85.4 | 27 MB | 6.5 M | - | - |
| PCNN [35] | CNN | - | 81.8 | 85.1 | 62 MB | 8.2 M | 326 ms | 84.2 |
| DGCNN [34] | Graph | 2.4 GB | - | 85.2 | 22 MB | 1.85 M | 340 ms | 84.8 |
| PT [36] | Trans | 8.2 GB | 82.6 | 85.8 | 51 MB | 15.3 M | - | - |
| PointNet++_MSG [26] | MLP | 2.0 GB | 81.9 | 85.1 | 26.8 MB | 1.7 M | 278 ms | 83.6 |
| PointNet++_SSG [26] | MLP | 1.8 GB | 81.2 | 84.7 | 24 MB | 1.5 M | 260 ms | 82.8 |
| PointNet [25] | MLP | 1.5 GB | 80.3 | 83.7 | 30.8 MB | 3.5 M | 156 ms | 82.1 |
| 3DSatNet | MLP | 1.5 GB | 82.1 | 85.3 | 20.8 MB | 1.3 M | 228 ms | 84.7 |

### 3.1.2. Overall Performance on 3D Spacecraft Component Segmentation Dataset

This section contains two groups of experiments. The first group experiments explore how to train the network, and the second group compares 3DSatNet network with SOTA networks in our dataset. All experiments in this section are tested on embedded computing devices.

The proposed 3D spacecraft component segmentation dataset is insufficient enough compared with the ShapeNet-Part dataset, which even cannot be improved despite we adopt multiple sensors to acquire the 3D point clouds. Training the 3DSatNet directly on our dataset takes the risk of over-fitting and affects the generalization of the model. Thus, we train a fine-tune model based on the spacecraft dataset and initialize the model weights with ones that are pre-trained on the ShapeNet-Part dataset. To be specific, we establish a pre-trained weights on the ShapeNet-Part dataset and then froze layers except for the FC layers. Only the FC layers are trained during the training process. The training process is run on the Nvidia RTX 3090 with 24 GB memory, and all testing process based on the proposed dataset is run on the Nvidia Jetson TX2. Nvidia Jetson TX2 https://www.nvidia.cn/autonomous-machines/embedded-systems/jetson-tx2/, accessed on 23 April 2022 is a power-efficient embedded 7.5-watt AI computing device. It's built around an NVIDIA Pascal™-family GPU and loaded with 8 GB of memory and 59.7 GB/s of memory bandwidth.

The first experiment demonstrates the influence of different training data on segmentation results through several comparative experiments. As shown in Table 5, considering that the data obtained on-orbit are actual point clouds, the test part only contains actual lidar point clouds and actual visual data. The training process requires large amounts of data and cannot be completed using only actual lidar or actual visual point clouds. Therefore, in the training data, the Lidar only in Table 5 means that the training part contains only simulated lidar and real lidar point clouds, while Lidar & Visual indicates that the training sample contains four kinds of point clouds.

**Table 5.** Studies on the different sources in 3D component segmentation dataset (%).

| Model | Train | Train Point Cloud Number | Test | Test Point Cloud Number | Instance mIoU |
|-------|-------|--------------------------|------|-------------------------|---------------|
| A | Lidar only | 366 | Actual Lidar | 30 | 84.2 |
| B | Lidar & Visual | 762 | Actual Lidar | 30 | 84.5 |
| C | Visual only | 366 | Actual Visual | 30 | 82.8 |
| D | Lidar & Visual | 762 | Actual Visual | 30 | 83.3 |

Compare experiment A with B, adding visual point clouds to the training part can improve the instance mIoU, experiment C and D can get the same conclusion. So we suggest using all sources of point cloud to train the fine-tuned 3DSatNet network.

Secondly, we try to deploy all the advanced algorithms in Table 4 on TX2. Unfortunately, KPconv [23], Point transformer [36], SpiderCNN [33], and other algorithms can not be deployed due to their high GPU memory usage and large model size.

This experiment compares the proposed algorithm with five networks: PointNet [25], PointNet++_SSG [26], PointNet++_MSG [26], PCNN [35], and DGCNN [34] for spacecraft segmentation on the proposed dataset.

The segmentation results are listed in Table 4. Although DGCNN achieves the best instance mIoU, the KNN computation and dynamic kernel computation are quite time-expensive. Our 3DSatNet achieves an IoU of 84.7%, which is superior to the other three MLP-based models. This experiment demonstrates that the ISS point selecting methods and the weighted cross-entropy loss proposed in our algorithm can improve the segmentation accuracy. Furthermore, the pruning of T-Net can make the model size reduced by 13.3% compared with PointNet++_SSG.

### 3.1.3. Components Segmentation Results Using 3DSatNet and 3DSatNet-Reg

In this section, we conduct an experiment to support our claim that the 3DSatNet-Reg has better performance than 3DSatNet. The experiment is based on a lidar point cloud generated by Arisim using the CAD model of Mars https://nasa3d.arc.nasa.gov/detail/eoss-mgs, accessed on 23 April 2022 with special components and a visual point cloud generated by visualSFM using the CAD model of Aqua https://nasa3d.arc.nasa.gov/detail/jpl-vtad-aqua, accessed on 23 April 2022. The visual point cloud contains a lot of noise. The visualization and precision analysis of the component segmentation results via 3DSatNet and 3DSatNet-Reg are presented in Figure 14 and Table 6.

Note that the 3D component segmentation results generated by other networks are listed in Figure A1.

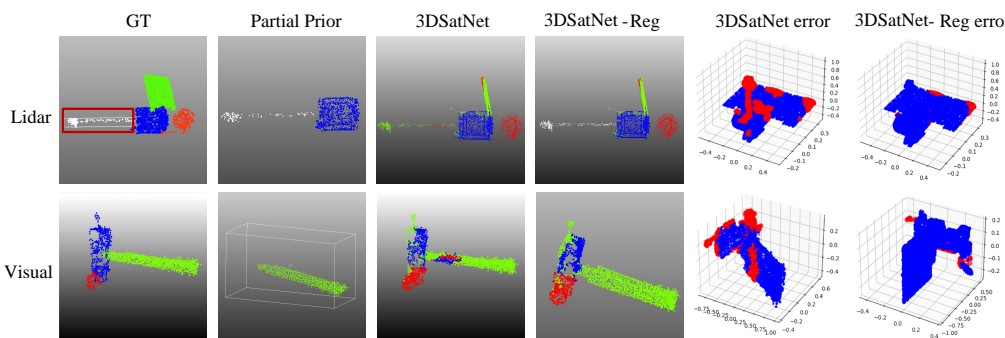

**Figure 14.** Visualization of the component segmentation results via different networks. (Blue: panel, Green: body, Red: Antenna, Grey: special components. In 3DSatNet error and 3DSatNet error Figures: red points represent wrong segmented points, blue points represent points that are segmented correctly).

**Table 6.** Accuracy evaluations of the component segmentation result via different models (%).

| Categories | Methods | Body | Panel | Antenna | Special |
|---|---|---|---|---|---|
| Lidar | DGCNN [34] | 87.1 | 84.2 | 68.2 | - |
| | PCNN [35] | 86.5 | 84.4 | 67.5 | - |
| | PointNet++(MSG) [26] | 85.3 | 83.4 | 62.5 | - |
| | PointNet++(SSG) [26] | 84.8 | 82.6 | 58.2 | - |
| | PointNet [25] | 84.3 | 81.5 | 56.1 | - |
| | 3DSatNet | 86.2 | 83.8 | 68.2 | - |
| | 3DSatNet-Reg | 86.2 | 83.7 | 68.2 | 89.2 |
| Visual | DGCNN [34] | 82.2 | 80.7 | 47.5 | - |
| | PCNN [35] | 81.2 | 80.2 | 43.8 | - |
| | PointNet++(MSG) [26] | 80.2 | 78.6 | 43.5 | - |
| | PointNet++(SSG) [26] | 79.3 | 78.4 | 38.4 | - |
| | PointNet [25] | 78.2 | 76.5 | 31.2 | - |
| | 3DSatNet | 81.3 | 80.9 | 48.5 | - |
| | 3DSatNet-Reg | 81.2 | 91.1 | 48.5 | - |

The lidar point cloud contains a special component that does not appear in the dataset, as the red box shown in lidar point clouds in Figure 14. Due to this inexistent component, 3DSatNet fails to generate semantic labels. However, when the prior model of the component is known, its semantic label can be predicted by the 3DSatNet-Reg and achieve an IoU of 89.2%.

Due to the presence of noise in the visual point cloud, the component segmentation results are far from expectations. The 3DSatNet network segments poorly even upon these popular components, such as body, panel. When the prior of the panel is provided, we can get the accurate segmentation result of the panel using registration.

The segmentation precision is shown in Table 6. Quantitative analysis shows that our 3DSatNet-Reg has achieved better results than the 3DSatNet, and can accurately generate the semantic labels of the components which are inexistent in the dataset using the supervision of registration.

### 3.2. Ablation

We design and perform extensive ablation studies on the proposed 3D spacecraft component segmentation dataset to analyze the significance of different components proposed for the 3DSatNet: the Fine Tune methods, the proposed geometrical-aware FE layer, and Our loss function.

We run tests on the 3D spacecraft component segmentation dataset under the same settings, and Table 7 presents the results. Model A means to train the 3DSatNet network on our dataset directly. Model B, C, and D use the weights trained on the ShapeNet-Part dataset as the weights of the trunk network, and only the FC layers and trained on our dataset. Model C uses the geometrical-aware FE layer, and model D uses both the FE layer and the proposed loss function.

As is shown in Table 7, directly training 3DSatNet network on our dataset has the problem of over-fitting. Considering there is only one class in our dataset, so we only compute the instance mIoU. Comparing model D to model B and model C, we observe that the proposed FE layer and the proposed loss function significantly improve the part segmentation's network performance. Compared to model B, model D's instance mIoU improved by 1.5%. These ablation experiments are further described in the following sections.

**Table 7.** Ablation studies on the proposed 3D component segmentation dataset (%).

| Model | Fine-Tune | Our Geometrical-Aware FE Layer | Our Loss Function | Instance mIoU |
|-------|-----------|-------------------------------|-------------------|---------------|
| A | - | - | - | over-fitting |
| B | ✓ | - | - | 83.2 |
| C | ✓ | ✓ | - | 84.0 |
| D | ✓ | ✓ | ✓ | 84.7 |

3.2.1. Comparisons on the Feature Extraction Methods

In this experiment, we compare the proposed ISS with FPS and random selection methods. The three point selection methods are used to select 5 to 150 points. The sampling results are then processed by 3DSatNet on the spacecraft dataset.

The component segmentation results using three point selection methods are shown in Figure 15. The points selected by FPS contain no structure information, which leads to the network performing badly on some spacecraft's components with few points.The ISS point selection method achieved the best performance. When the number of selected key points is 20, the performance of the ISS-based point selection method achieved 11% higher accuracy than the FPS. With the increase of sampling number, the performance of the ISS is slightly better than FPS and a random selection, the advantage of the proposed method became less evident. That is because, with a large sampling numbers, randomly selected and FPS can ultimately sample the points along the spacecraft edges. The experiment results demonstrate that if the reconstructed point cloud is sparse, for example, the total number is below 150, it is crucial to use the ISS point selection method for a better component segmentation accuracy.

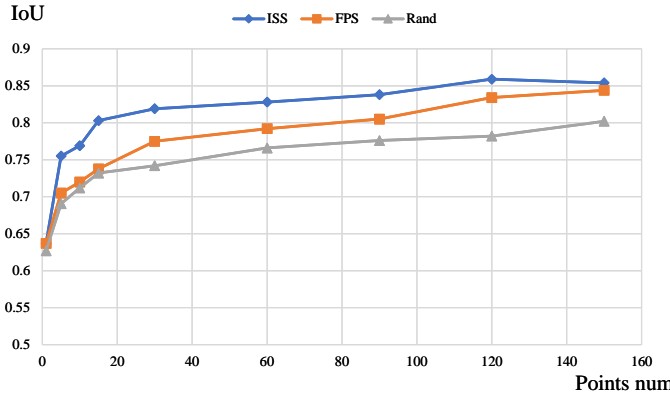

**Figure 15.** Component segmentation accuracy of three point selection methods.

3.2.2. Comparisons on the Different Loss Functions

This experiment tests the different component segmentation performances using two different losses of point clouds: the proposed loss function and cross-entropy loss. The component segmentation visualization is shown in Figure 16. The accuracy and IoU with are defined the same as [25] are listed in Table 8. The red dots represent the points predicted incorrectly, and the blue dots represent the points predicted correctly.

It can be seen in Figure 16 and Table 8 that while using the proposed loss function, the component segmentation accuracy of body, panel, and antenna are all increased. Particularly, the IoU of the antenna significantly increase by 15.8% while the segmentation IoU of other categories not be affected.

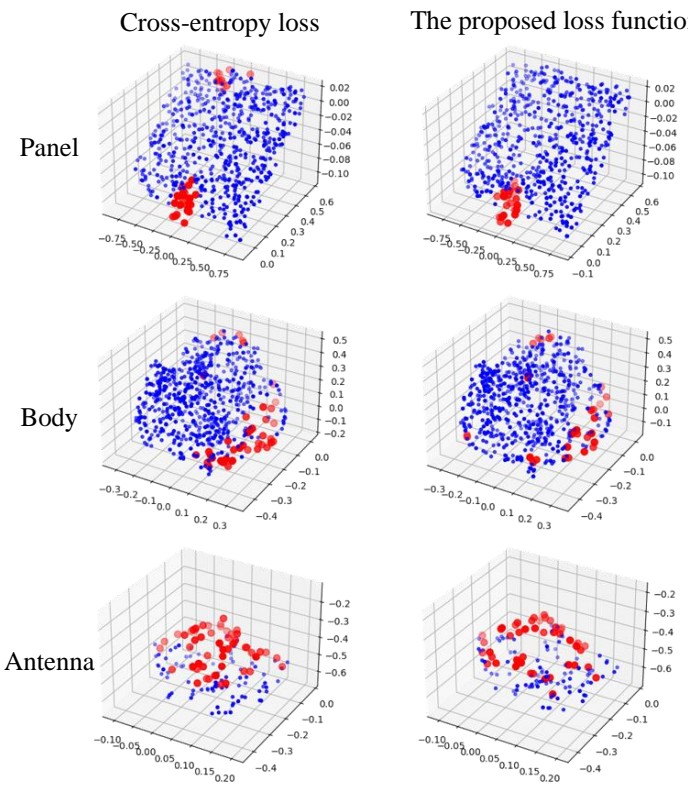

**Figure 16.** The segmentation error visualization of cross-entropy and the proposed loss function. (red: wrong segmented points; blue: correct segmented points).

**Table 8.** Accuracy evaluations of cross-entropy loss and the proposed loss function (%).

| Components | Methods | Accuracy | IoU |
|---|---|---|---|
| Body | Cross-entropy loss | 89.1 | 80.3 |
| | The proposed loss function | 91.2 | 82.1 |
| Panel | Cross-entropy loss | 84.6 | 80.7 |
| | The proposed loss function | 92.3 | 84.6 |
| Antenna | Cross-entropy loss | 95.2 | 41.8 |
| | The proposed loss function | 93.1 | 57.6 |

### 3.2.3. Registration Results

This section verifies the feasibility of the registration scheme proposed in this paper. Experiments are carried out on two types of point clouds: visual point clouds and lidar point clouds. The experimental results are shown in Figure 17.

As shown in Figure 17, the first two columns are the results of D3Feat feature points (red points shown in Figure 17) extracted from the global point cloud and the prior point clouds respectively. The third column demonstrates the initial poses of the prior point clouds and the semantic point clouds gained by 3DSatNet, and their registration results are depicted in the fourth column. Further, the fifth column visualized the accuracy evaluation results of registration, and the sixth column presents the diagram of average precision. A conclusion can be drawn from Figure 17 that the proposed 3DSatNet-Reg can correctly register the two point clouds.

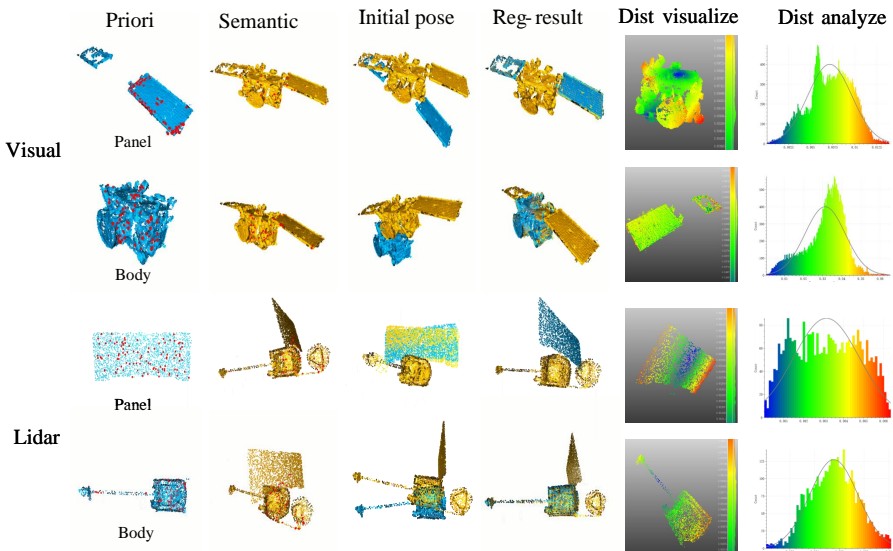

**Figure 17.** Partial prior point clouds registered with semantic point clouds generated by 3DSatNet.

We calculate the Euclidean Distance of the nearest points between the partial point cloud $S_1$ and the complete point cloud $S_2$ in the accuracy evaluation and utilize average distance to present the registration error. It is defined as:

$$d(x, S_2) = \min_{y \in S_2} \sqrt{(x - y)^2} \tag{13}$$

As shown in the sixth column in Figure 17, for the visual point cloud with the size of 1.25 m $\times$ 0.56 m $\times$ 0.48 m, the average registration accuracy of the body part is 0.7 cm, while it is 3.2 cm for the panel due to the presence of noise. As for the simulated lidar point cloud with the size of 1.35 m $\times$ 0.86 m $\times$ 0.65 m, the average registration accuracy of the body part is 0.56 cm, and that is 0.51 cm for the panel with less noise.

As we can see in Table 9, registering partial prior point clouds with the semantic point clouds gained by 3DSatNet cannot be achieved by ICP [31] alone. Comparing model B to model C and model D to model E, adding ICP [31] behind Teaser [28] can obviously improve the registration accuracy. Using Teaser's registration results as the initial transformation, ICP only needs a small number of iterations to get nearly perfect registration results. Comparing model C to model E, the deep learning-based D3Feat [30] can obtain repeatable keypoints for 3D point clouds compared with FPFH [37].

**Table 9.** Accuracy evaluations of registration.

| Model | Feature | Registration Methods | Lidar Point Clouds | | Visual Point Clouds | |
| --- | --- | --- | --- | --- | --- | --- |
| | | | Error | Time | Error | Time |
| A | - | ICP [31] | - | - | - | - |
| B | FPFH [37] | Teaser [28] | 1.02 cm | 0.68 s | 7.28 cm | 0.75 s |
| C | FPFH [37] | Teaser + ICP | 0.67 cm | 0.86 s | 6.05 cm | 1.24 s |
| D | D3Feat [30] | Teaser [28] | 0.96 cm | 0.74 s | 6.35 cm | 1.02 s |
| E | D3Feat [30] | Teaser + ICP | 0.43 cm | 0.99 s | 5.14 cm | 1.46 s |

## 4. Discussion

This section describes a few limitations and planned future improvements on our 3DSatNet performance and 3D spacecraft component segmentation dataset.

*4.1. Dataset Limitations*

The proposed 3D spacecraft component segmentation dataset is insufficient enough to train large scale networks, such as Kpconv [23], CapsuleNet [38], DGCNN [34], and Point Transformer [36]. To solve this problem, firstly, we have adopted multiple sensors to acquire 3D point clouds. Then, we initialize the model weights with ones that are pre-trained on the ShapeNet-Part dataset. Therefore, future sequels of the dataset will include more spacecraft to facilitate the study of deep learning techniques beyond the application of 3D component segmentation about non-cooperative spacecraft. Furthermore, more advanced point cloud filtering methods can be used to improve the quality of the visual point clouds. For example, we can adopt GAN (Generative Adversarial Networks) to complete the point clouds.

*4.2. Relation to Real Mission Constraints*

In real missions, lidar is used to compute distances and relative poses. When the distance is long, the spacecraft contains only a few points, which cannot conduct component segmentation. For practical application, it is necessary to perform a fly-around task before component segmentation, using lidar odometry to reconstruct complete spacecraft point clouds, or using visual SLAM reconstruction to obtain sparse point clouds. In 3DSatNet-Reg, the prior point clouds obtained are not complete, and the density of the prior point clouds and semantic point clouds is also different. Future work will use some point clouds from lidar odometry to verify the registration accuracy.

*4.3. Further Refinement of Segmentation Accuracy of Components with Fewer Points*

To find a 3D component segmentation network suitable for spacecraft, our methods are based on the PointNet++ network and contain many improvements. A new weighted cross-entropy loss is proposed to increase the weight of components with fewer points. The pruning operation is performed on the model to reduce the size of 3DSatNet. The effect of improvement is still insignificant due to the limitation of MLP characterization capability. It is necessary to explore lightweight networks and introduce attention mechanisms to achieve higher-precision 3D spacecraft component segmentation results. Moreover, the accuracy of semantic correction in 3DSatNet-Reg depends on the accuracy of registration, more efforts should be made to increase registration accuracy.

**5. Conclusions**

This paper proposes a 3D spacecraft component segmentation dataset with multi-source spacecraft point clouds. Then, A 3D spacecraft component segmentation network named 3DSatNet is proposed with a geometrical-aware FE layer to extract geometric and transformation invariant features and a new loss function to increase the segmentation performance of spacecraft components with fewer points. Experiments carried out using the proposed dataset and the ShapeNet prove that our network achieves better segmentation instance mIoU compared to all MLP-Based networks, and achieves the smaller forward time and model size compared to CNN-based, Graph-based networks. Specifically, when the prior point clouds are known, 3DSatNet-Reg is recommended. By registering the labeled prior point clouds with the on-orbit semantic point cloud, 3DSatNet-Reg has the ability to segment special components that are not contained in the dataset.

**Author Contributions:** Conceptualization, G.Z. and X.W.; Data curation, G.Z.; Formal analysis, G.Z., X.W., Y.T. and Y.S.; Funding acquisition, X.W.; Methodology, G.Z. and X.W.; Project administration, X.W. and S.L.; Resources, G.Z. and X.W.; Supervision, X.W. and S.L.; Writing – original draft, G.Z., X.W., Y.T. and Y.S.; writing—review and editing, G.Z., X.W. and Y.T. All authors have read and agreed to the published version of the manuscript.

**Funding:** This research was funded by Chinese Academy of Sciences (No.Y9030971WY).

**Institutional Review Board Statement:** Not applicable.

**Informed Consent Statement:** Not applicable.

**Data Availability Statement:** The 3D CAD models of spacecraft are available at https://nasa3d.arc.nasa.gov/, accessed on 26 April 2022. Everyone can access partial of the proposed 3D spacecraft component segmentation dataset by visiting our project home page: https://github.com/GY-ZHAO/3D_spacecraft_component_segmentation_dataset, accessed on 26 April 2022. Our code and some spacecraft component segmentation results are published in https://github.com/GY-ZHAO/3DSatNet, accessed on 23 April 2022.

**Acknowledgments:** We thank NASA for providing CAD models of spacecraft online.

**Conflicts of Interest:** The authors declare no conflict of interest.

## Abbreviations

The following abbreviations are used in this manuscript:

| | |
|---|---|
| OOS | On-Orbit Service |
| DEOS | Deutsche Orbital Servicing Mission |
| iBOSS | Intelligent Building Blocks for On-Orbit Satellite Servicing |
| SIS | Space Infrastructure Servicing |
| NASA | National Aeronautics and Space Administration |
| FOV | Field of View |
| SGM | Semi-Global Matching |
| RANSAC | RANdom SAmple Consensus |
| ToF | Time of Flight |
| ICP | Iterative Closest Point |
| MLP | Multi-layer Perception |
| AI | Artificial Intelligence |
| SLAM | Simultaneous Localization and Mapping |
| ISS | Intrinsic Shape Signatures |
| EVD | Eigenvalue Decomposition |
| SOTA | State Of The Art |
| FPS | Farthest Point Selection |
| TLS | Truncated Least Squares |
| TRIMS | Translation and Rotation Invariant Measurements |
| GAN | Generative Adversarial Networks |
| GT | Ground Truth |

## Appendix A

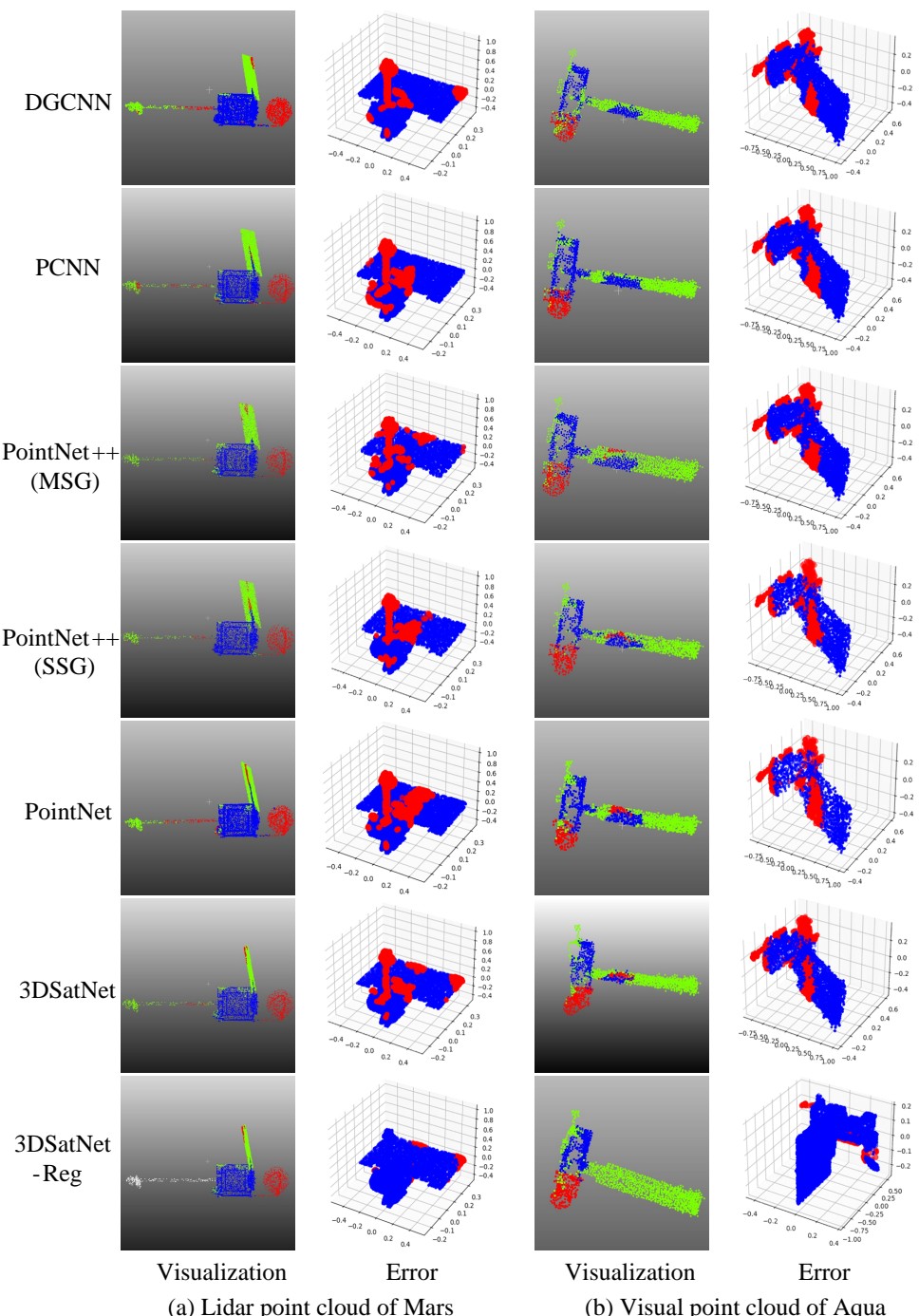

**Figure A1.** Spacecraft component segmentation results using special networks. (Blue: panel, Green: body, Red: Antenna. In error Figures: red points represent wrong segmented points, blue points represent points that are segmented correctly).

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
