# Peer review of "3D Component Segmentation Network and Dataset for Non-Cooperative Spacecraft"

_aerospace, doi:10.3390/aerospace9050248_

Round 1

Reviewer 1 Report

The article "3D Component Segmentation Network for Non-cooperative Spacecraft" presents an innovative method of segmentation of elements (devices) of ships flying in orbit. The work is well structured and clear. I have posted my comments below.

  1. Who built the 3DSatNet network? Is it a product of your project?
  2. equation (1): from mathematic point of view this is not precise. Can you describe its details? max(P) - what is it?
  3. line 186: Please check how to refer to material from the Internet.
  4. line 109, 124: Teaser: references must be added here
  5. equation 2: the sum symbol is misleading, I suggest you change to a different one
  6. equation 2: all the parameters and variables must be explained below the equation
  7. line 216: Given - change to given
  8. equation 4: "Nc represents the ...." - where is Nc?Please check carefully all the formulas in the article and whether each of the symbols used is described.

Reviewer 2 Report

1. For readers to quickly catch the contribution in this work, it would be better to highlight major difficulties and challenges, and your original achievements to overcome them, in a clearer way in abstract and introduction. 

2. It is suggested to further highlight the challenges of the investigated problem.

3. More recent works can be discussed in Introduction and the paper should be linked to literature of the topic addressed, particularly on fault detection approaches, influence of disturbances, modeling errors, various uncertainties in the real systems. A relevant recent review are: Iterative learning control for repetitive tasks with randomly varying trial lengths using successive projection, International Journal of Adaptive Control and Signal Processing; Asynchronous Fault Detection Observer for 2-D Markov Jump Systems, IEEE Transactions on Cybernetics; It is necessary to comment what would be changed in this case and make relation with the papers on this topic in Introduction section, and in that way, point out other contemporary approaches and possibilities. I believe this would further strengthen the introduction and lend support to the methodology applied in general.

4. Authors should argue their choice of the performance evaluation indicators.

5. How about the complexity of the proposed approach? Please give some analysis.

Reviewer 3 Report

Summary

The work entitled “3D Component Segmentation Network for Non-cooperative Spacecraft” presents a dataset based on 3D point clouds for training deep learning models on spacecraft component segmentation, as well as two network architectures aimed at improving component segmentation accuracy-latency trade-off on space representative hardware with respect to the state-of-the-art.

After an overview of the methods employed to generate a multi-source 3D spacecraft component segmentation dataset, the authors describe a new network architecture based on PointNet. The main novelty of the network lies in a new feature-extraction layer which makes this task robust to rigid geometric transformations. A new training loss is introduced to deal with the problem that small components are difficult to learn as they are represented by a small fraction of the point cloud. They refer to this as the data imbalance problem. Finally, a registration method is embedded into the model to increase segmentation accuracy when partial prior point clouds are available.

Reviewer’s recommendation

Overall, the work contains several strengths. First, the dataset proposed is the first of its kind focusing on space targets. Second, the models have been developed with a focus on memory footprint and computational burden which play an essential role in real time applicability on space graded hardware. A comparison with the state-of-the-art and an ablation study to investigate the effectiveness of each contribution are reported in section 3. The limitations of the work have been well reported by the authors in section 4, apart from the one raised in the third paragraph of the section “Major issues”.

In light of that, I recommend only minor adjustments to the manuscript, as detailed in the following.

Major Issues

Introduction: in my opinion the first part of the literature review is not well structured. After a general introduction to OOS in the first paragraph, the authors move the attention to the topic of component level segmentation (line 31), but many of the papers mentioned hereafter [13-16] are not fully relevant to that topic, as it can be inferred by the summary they provide (lines 37-44). Then, line 47 contains another statement focusing again the attention on component recognition and localization followed by a summary of relevant works. This suggests that the purpose of the authors was first to demonstrate how deep learning is spreading out in the space field and then present works related to the topics of the paper. In this case, for the first part I would suggest to defer the reader to a review paper (e.g. The Final Frontier: Deep Learning in Space) also because the few references mentioned are not representative of recent advancements and a thorough review is out of the scope of the work.

In line 35 the authors claim that “Several datasets, such as SPARK[10], SPEED[11], Spacecraft-Parts[12], have been published to evaluate the performance of deep learning algorithms in spacecraft 2D component segmentation”. This statement is inaccurate as only the last one has been developed for this purpose while the others lack information to perform component segmentation since they target different problems: spacecraft recognition and 6D pose estimation, respectively. This is clearly stated in reference [10] Table 1.

Section 2, the point clouds obtained from the three different sources are mixed together to get a rich representation of the target satellites. While on ground one might have access to all the sources, which is fine to get more data for training the deep learning model, a satellite will most likely carry only one of such sensors and technologies. These might have different accuracies (for example the authors mention manual removal of noise in line 164) and might return a different level of discretization of the target. Therefore, the results presented might not be representative of an in-orbit scenario since it is not possible to distinguish the performance of the method on each source. I suggest to include in the dataset a label identifying the source (Fig. 9) and to test the model on each of them individually.

Minor Issues

The title is only partially representative of the contents as it does not mention the creation of a new dataset. Being the first of its kind it would be worth to emphasize it also in the title.

Abstract: line 8 it is not clear to what the “data imbalance problem” refers to, which is explained only in line 88.

Line 45, “higher intelligent navigation” might be replaced with something like “more complex navigation related tasks”.

Section 2.1, it is not easy to follow how many data have been collected from the different sources. For example, I suppose VisualSFM has been applied on the images of all the 28 CAD models (collected on Blender) and on the ones obtained from the Nvidia Jetson Camera navigating around the 5 3D printed models. I suggest to include a table summarizing these information.

Lines 131-132 may be skipped as the topics have already been summarized at the end of the introduction and detailed at the beginning of section 2.

In many parts of the paper (e.g. lines 134, 135, 153, 158) the authors inserted references and access dates to resources other than papers. I suggest to include them into the refences as well, or to move them in footnotes for readability.

Figure 3, the graphs on the left miss labels and units.

Line 157, it is not clear how the authors navigated the Nvidia Jetson TX2 camera around the target.

Lines 225-226: from my understanding by removing the T-Net the authors allow the point clouds entering the PointNet to have different possible view angles, scales and shifts. They tackle this issue with the new Feature Extraction layer. However, from these lines it is not fully clear what they mean and shall be re-phrased: is the meaning that this re-alignment is “learnt” by the MLP?
